# Progression in training volume and perceived psychological and physiological training distress in Norwegian student athletes: A cross-sectional study

**Cathrine Nyhus Hagum**[1]*, **Espen Tønnessen**[2], **Shaher A. I. Shalfawi**[1]

**1** Department of Education and Sports Science, University of Stavanger, Stavanger, Norway, **2** Faculty of Health Sciences, Kristiania University College, Oslo, Norway

* cathrine.n.hagum@uis.no

**Data Availability Statement:** All relevant data are within the paper and its Supporting Information files.

## Abstract

This cross-sectional study examined self-reported weekly training volume and perceived training distress in Norwegian student athletes according to gender, type of sport, school program, and school year. The Norwegian version of the Multicomponent Training Distress Scale (MTDS-N) was completed by 608 student athletes (*M* age = 17.29 ± .94). Univariate and multivariate techniques were used in data analyses. Results revealed significant differences in weekly training volume between sport types. No significant differences in weekly training volume were found for gender, school year, or school program. However, a multivariate effect was found for gender, with females perceiving higher levels of training distress than males. A multivariate interaction effect between school year and training volume was also observed. We recommend that practitioners use a conceptual framework to periodize training and monitor training distress in student athletes, particularly in females, to preserve physiological and psychological well-being and ensure a progressive training overload leading to positive performance development.

## Introduction

Becoming a world-class athlete requires systematic, quality training over time [1]. Data on elite female and male athletes from different sports indicate that athletes with an average of 10.5 training years have five training sessions and 16 hours of training per week with ~2.5 hours per training session and approximately 18 competitions a year [1]. The quality of the training is influenced by the training prescription, which should be in line with the desired outcome (i.e., goal/s), and is defined in terms of training volume, intensity, and frequency [2]. Research shows that these three components collectively referred to as training load, influence training adaptation and prevent or cause overtraining, illness, and injury [3]. Therefore, the optimal training outcomes depend on an adequate balance between training load components and non-training loads (i.e., stressors) and recovery [4, 5]. Hence, ongoing monitoring and modification of these elements are crucial in developing an optimal training prescription that can lead to high-standard performance and minimize undesired training outcomes [6–9].

**Funding:** The author(s) received no specific funding for this work.

**Competing interests:** The authors have declared that no competing interests exist.

When determining the type and amount of training necessary at different stages of an athletic career, it is critical to understand the physiological and psychological demands arising from both the sport (i.e., its physiological and biomechanical profile) and the athlete's developmental stage [10, 11]. For example, puberty can be challenging in athletes' careers, with significant hormonally driven physical changes occurring in males and females, causing the body to respond differently to exercise. In addition, a rapid increase in growth has also been associated with an increased risk of bone and growth plate injuries [12]. Moreover, puberty can often be psychologically challenging, especially for females [13]. Another potentially challenging period is the transition from the lower secondary to the upper secondary school, which typically involves an increased training load [14–16] combined with school and other life demands [17]. Hence, both boys and girls can experience tremendous psychological pressure during this phase [17].

Understanding the sport's demands and the different stages in an athlete's development can help determine the optimal magnitude of the training components to target the desired outcome (i.e., goal/s) at different stages in an athlete's career. Practitioners can then monitor how athletes tolerate training load and make the necessary adjustments to optimize the physiological performance capacity [7, 18]. Furthermore, reference values can be established regarding training volumes in different sports and recommended progression from year to year, making it easier for both coaches and athletes to design optimal training plans. For example, elite athletes complete between 800–1200 training hours per year in typical endurance sports such as cross-country skiing [19–22], rowing [23, 24], triathlon [25], and swimming [1]. In more technically demanding sports such as soccer [26], handball [27], and athletics [28, 29], elite athletes complete between 500–700 annual training hours.

Several tools have been developed to monitor athletes' physical internal and external training loads [9, 18]. However, a holistic approach to athletes' monitoring should be adopted to consider physiological and psychological factors, especially for younger athletes with significant physiological and lifestyle changes [30]. Hence, the Multicomponent Training Distress Scale (MTDS) is a simple athlete self-report measure that combines physical and psychological stressors [31]. The questionnaire has been translated into Norwegian and assessed for its factorial validity. However, the relationship between physical and psychological training distress and different characteristics in student athletes in Norway has not been elucidated [32]. Therefore, the dual aims of this study were:

1. To describe student athletes´ weekly training volume in Norwegian upper secondary schools and determine differences in training volume according to gender, type of sport, school program, and school year.

2. To investigate whether weekly training volume, gender, type of sport, school program, or school year influence responses to the dimensions in the Norwegian Multicomponent Training Distress Scale (MTDS-N) and whether there are any interaction effects between these variables.

We had two general pre-specified research questions that we aimed to answer:

**Question 1a:** Are there any differences in training volume according to the type of sport?

**Question 1b:** Are there any differences in training volume according to the school program (i.e., students attending sports and physical education versus students attending specialization in general studies)?

**Question 1c:** Are there any differences in training volume according to school year (i.e., first, second and third-year students)?

**Question 2a:** Does the weekly training volume influence the responses to the dimensions in MTDS-N?

**Question 2b:** Does gender influences responses to the dimensions in MTDS-N?

## Materials and methods

### Participants

The "point of stability" approach was used to estimate the sample size [33–35]. This approach ensures that the deviation between the estimated sample and the population parameter is stable (small) and predicted to remain small at a stable statistical power (80%) [33, 34]. According to Cohen [36], to ensure small stability, the corridor of stability should not exceed a small correlation of 0.10. Schönbrodt and Perugini [34] suggested that the minimum number needed to reach the point of stability would be 240–250 participants. According to Kretzschmar and Gignac [33] the point-estimates of the correlation was stabilized at a sample size of 220 with perfect reliability (omega, $\omega = 1.0$) of both latent factors and a population correlation of $p = 0.20$. Because perfect reliability is almost never attained, the authors proposed that the required sample at a population correlation of $p = 0.20$ and reliability of $\omega = 0.7$ would be $\geq$ 490 participants [33]. Hirschfeld, Brachel and Thielsch [35] have reported similar results with the recommended sample size to reach a point of stability was > 500 participants. Consequently, the sample size that was required in this study was to be more or equal to the recommendations from comparable studies (i.e., $n \geq 500$).

The participants ($n = 632$) were recruited from 34 Norwegian upper secondary schools offering the optional subject "top-standard sport." This study was conducted according to the principles expressed in the Declaration of Helsinki, and all participants provided their written, informed consent. Furthermore, the study was approved by the Norwegian Social Science Data Services (NSD) (Project number: 836079) and the Regional Committees for Medical and Health Research Ethics (REK) (project number: 54584). Participants reporting $\leq$ 4 hours of training per week (n = 21) were excluded from the data analysis to guarantee a minimum training volume. Further, outliers in preliminary analyses with $\geq$ 30 hours of training per week ($n = 3$) were excluded, leaving a total sample size of 608 student athletes (308 male, 298 female, $M$ age = 17.29 ± .94 years). The student athletes participated in a range of team ($n = 405$; e.g., soccer) and individual ($n = 202$; e.g., athletics) sports, training on average 12.76 hours (± 4.45) per week.

### Instruments and procedures

The MTDS questionnaire was used to assess and describe the student athletes´ training distress [31]. The instrument consists of 22 items and six factors (depression, vigour, physical symptoms, sleep disturbances, stress, and fatigue). Depression, vigour, stress, and fatigue are measured in terms of their frequency and scored on a five-point Likert scale ranging from "never" (1)–"very often" (5). Physical symptoms and sleep disturbances are measured in terms of their intensity and scored on a five-point Likert scale ranging from "not at all" (1)–"an extreme amount" (5). Before data collection, the questionnaire was translated into Norwegian and assessed for factorial validity [32]. All upper secondary schools that offer the optional program subject top-standard sport in Norway ($n = 119$) were invited to participate in the present study. The MTDS-N was distributed electronically using SurveyXact version 8.0 [37] to the school management who agreed to participate ($n = 34$, 28.6%). Further, the school management distributed the questionnaire electronically to the student athletes at their respective schools ($n = 23$, 19.3%). The data collection took place during class and started in March 2020

and ended in May 2020. To assess the student athletes' training volume, student athletes reported their current weekly training hours. In addition, the survey included questions regarding age, gender, county, school name, study program, school year, and primary type of sport. The instrument and data collection procedure are fully described in [32].

Data analyses

All analyses were carried out using Statistical Package for the Social Sciences (SPSS) Version 25 (IBM Corporation, Armonk, NY, USA). First, the factor vigour from the MTDS questionnaire, with positive scores, was reversed. Descriptive statistics for all variables are presented as mean (*M*) and standard deviation of the mean (*SD*). Then, to investigate the difference in weekly training volume according to gender, type of sport, school program, and school year (independent variables), multiple one-way analyses of variance (ANOVA) were conducted. A Bonferroni adjustment was applied to correct for multiple comparisons and reduce the likelihood of Type I error [38, 39]. Next, a two-way ANOVA was conducted to investigate the trend in weekly training volume across the three school years and different sport types. Partial eta squared ($\eta_p^2$) was used to determine the effect size and were interpreted as 0.01 = small, 0.06 = medium, or 0.14 = large [36]. To assess whether the independent variables influenced the dependent variables in MTDS-N (i.e., depression, vigour, sleep disturbances, physical symptoms, stress, and fatigue), or if there was an interaction between training volume and the independent variables, four different factorial multivariate analyses of variance (MANOVA) were conducted [40]. Before performing the MANOVAs, preliminary assumptions were assessed (i.e., correlations among the dependent variables, normality, outliers, and the homogeneity of variance-covariance matrices). The results of the preliminary assumptions met the criteria for running MANOVA (S1 Table; S1 File).

The first MANOVA had a 3×2 factorial design with weekly training volume (5–10 hours, 10–15 hours, $\geq$ 15 hours) and gender (males, females) as the independent variables. Cutpoints of 5, 10, and 15 hours of training per week were chosen to ensure relatively equal group sizes [41]. The second MANOVA had a 3×3 factorial design with weekly training volume and school year (first year, second year, third year) as the independent variables. The third MANOVA included weekly training volume and sports type (soccer, other team- and ball sports, endurance sports, weight-bearing sports, other sports; S2 Table) as the independent variables, resulting in a 3×5 factorial design. The fourth MANOVA consisted of weekly training volume and school program (specialization in general studies, sports and physical education), resulting in a 3×2 factorial design. The Wilks' lambda ($\lambda$) criterion was used to interpret the results of the MANOVA. However, if the Box's M test was statistically significant ($p < 0.001$), the Pillai's Trace was used to interpret the results of the MANOVA. The Pillai's Trace is considered a robust test in place of Wilk's Lambda if the assumption of homogeneity of variance-covariance matrices is violated [42, 43]. Furthermore, descriptive discriminant analysis (DDA) was conducted as a multivariate post-hoc analysis for evaluating the MANOVA effects, which has been recommended rather than running several ANOVAs to test mean differences [40, 44, 45]. The composite variable means (i.e., training distress) were used to examine differences between groups. If a statistically significant main effect was observed for an independent variable, a one-way ANOVA was conducted with either training volume, gender, type of sport, school year, sports type, or school program as the independent variable and the saved discriminate function scores as the dependent variable to determine the magnitude of group differences. Furthermore, to determine which groups differed on the interaction composite, a two-way ANOVA with Bonferroni adjustment was conducted when a statistically significant interaction effect was observed. Then, a multivariate interaction composite was created, which was used as the dependent variable. Cohen's *d* effect sizes were calculated using the composite variable means and *SD* of the groups on the composite dependent variable to examine the

composite variable means differences' magnitude and practical meaning [46]. Cohen's $d$ effect sizes were converted to Person's $r$ using Cohen's approximate conversion formula to measure the relationship between variables, and $r$ were then multiplied to the power of 2 (i.e., $r^2$) to be able to estimate the "variance-accounted-for" between variable [46]. The relationships between the variables were interpreted based on the guidelines proposed by Funder and Ozer [47], where an $r$ of 0.05 indicated a very small relationship; an $r$ of 0.10 indicated a small relationship; an $r$ of 0.20 indicated a medium relationship; an $r$ of 0.30 indicated a large relationship; and an $r$ of $\geq$ 0.40 indicated a very large relationship.

# Results

## Description of weekly training volume

Descriptive characteristics of the participants are presented in Table 1. Weekly training volume according to gender, type of sport, school program, and school year are presented in Table 2. The one-way ANOVA yielded a statistically significant difference in weekly training volume between the five sport types [$F_{(4, 588)}$ = 18.83, $p < 0.001$. The post-hoc test using Bonferroni adjustment indicated that student athletes playing soccer had a significantly less volume of training (11.69 hours $\pm$ 3.84) compared to those in endurance sports (15.06 hours $\pm$ 4.92; $M$ difference = -3.37 hours, $p < 0.001$, $d = 0.76$, $r = 0.36$), weight-bearing sports (14.56 hours $\pm$ 4.74; $M$ difference = -2.87 hours, $p < 0.001$, $d = 0.67$, $r = 0.32$), and other sports (15.10 $\pm$ 5.02; $M$ difference = -3.41 hours, $p < 0.001$, $d = 0.76$, $r = 0.36$). No significant differences in weekly training volume were found between soccer and other team- and ball sports

**Table 1. Descriptive characteristics of the 608 student athletes in the present study.**

| Characteristics (total)[1] | Modalities | Frequency or $M \pm SD$ | % |
|---|---|---|---|
| Gender (606) | Male | 308 | 50.8 |
| | Female | 298 | 49.2 |
| Age in years (yr) and months (mo) (607) | Total | 17 yr 3.5 mo $\pm$ 11.3 mo | |
| | Male | 17 yr 4.3 mo $\pm$ 11.5 mo | |
| | Female | 17 yr 2.6 mo $\pm$ 10.9 mo | |
| Region (608) | West Norway | 333 | 54.8 |
| | East Norway | 140 | 23.0 |
| | Mid Norway | 102 | 16.8 |
| | Northern Norway | 33 | 5.4 |
| School program[2] (608) | Specialization in general studies | 358 | 58.9 |
| | Sports and physical education | 250 | 41.1 |
| School year (608) | First year | 225 | 37.0 |
| | Second year | 234 | 38.5 |
| | Third year | 149 | 24.5 |
| Type of sport (607) | Soccer | 290 | 47.8 |
| | Other teams- and ball sports | 124 | 20.4 |
| | Endurance | 94 | 15.5 |
| | Weight-bearing sports | 52 | 8.6 |
| | Other sports | 47 | 7.7 |

$M$ = Mean; $SD$ = Standard Deviation; % = Percentage.

[1]Values in brackets indicate total responses from the participants.

[2] In specialization in general studies with top-standard sport, the student athletes attend regular specialization in general studies with the top-standard sport as an optional program subject. In sports and physical education, student athletes have theoretical and practical subjects related to sports. These include physical activity, sports science, training management, sports and society, and top-standard sport's optional program.

**Table 2. Descriptive statistics of weekly training volume[a] for gender, type of sport, school program, and school year.**

| Variable | | $n$ | $M$/h | $SD$/h | 95% CI/h | |
|---|---|---|---|---|---|---|
| Gender[b] | Male | 297 | 12.95 | 4.62 | 12.42 | 13.48 |
| | Female | 294 | 12.57 | 4.28 | 12.08 | 13.06 |
| | Total | 591 | 12.76 | 4.46 | 12.40 | 13.12 |
| Type of sport | Soccer | 283 | 11.69 | 3.84 | 11.24 | 12.14 |
| | Other teams- and ball sports | 120 | 11.85 | 3.84 | 11.16 | 12.54 |
| | Endurance | 93 | 15.06 | 4.92 | 14.05 | 16.07 |
| | Weight-bearing sports | 51 | 14.56 | 4.74 | 13.23 | 15.89 |
| | Other sports | 46 | 15.10 | 5.02 | 13.61 | 16.59 |
| | Total | 593 | 12.76 | 4.45 | 12.40 | 13.12 |
| School program | SGS | 351 | 12.69 | 4.37 | 12.23 | 13.15 |
| | SPE | 242 | 12.86 | 4.57 | 12.28 | 13.44 |
| | Total | 593 | 12.76 | 4.45 | 12.40 | 13.12 |
| School year | First year | 219 | 13.20 | 4.56 | 12.60 | 13.81 |
| | Second year | 229 | 12.58 | 4.11 | 12.05 | 13.12 |
| | Third year | 145 | 12.38 | 4.77 | 11.60 | 13.16 |
| | Total | 593 | 12.76 | 4.45 | 12.40 | 13.12 |

SGS = Specialization in general studies; SPE = Sports and physical education; n = sample size; M = mean; SD = standard deviation; CI = Confidence interval; h = hours.

[a] 15 missing values were observed for training volume.

[b] 2 missing values were observed for gender.

(11.85 hours ± 3.84; $M$ difference = -0.16 hours, $p$ = 1.000). Furthermore, student athletes in other team- and ball sports had a significantly less training volume compared to those in endurance sports ($M$ difference = -3.21 hours, $p < 0.001$, $d$ = 0.73, $r$ = 0.34), weight-bearing sports ($M$ difference = -2.71 hours, $p$ = 0.001, $d$ = 0.63, $r$ = 0.30), and other sports ($M$ difference = -3.25 hours, $p < 0.001$, $d$ = 0.73, $r$ = 0.34). No significant differences in weekly training volume were observed for gender [$F_{(1,589)}$ = 1.08, $p$ = 0.229], school program [$F_{(1,591)}$ = 0.20, $p$ = 0.652], or school year [$F_{(2,590)}$ = 1.80, $p$ = 0.166].

The two-way ANOVA indicated a statistically significant interaction between school year and sport type on weekly training volume [$F_{(8, 578)}$ = 1.978, $p$ = 0.047, $\eta_p^2$ = 0.027]. Simple main effects analysis showed no significant difference in weekly training volume across the three school years for soccer, other teams- and ball sports, or endurance sports. Student athletes in weight-bearing sports had a significantly less training volume in third year compared to first year ($M$ difference -4.04, $p$ = 0.020, $d$ = 0.81, $r$ = 0.38). Student athletes in other sports had a significantly larger training volume in third year compared to first year ($M$ difference 3.69, $p$ = 0.16, $d$ = 0.77, $r$ = 0.36) and second year ($M$ difference 3.58, $p$ = 0.03, $d$ = 0.71, $r$ = 0.34). Fig 1 illustrates weekly training volume across the school years for the five different sport types.

### Description of perceived psychological and physiological training distress

Table 3 reports descriptive statistics of the six dimensions of MTDS-N for male and female student athletes.

### The effect of training volume, gender, school year, sport types, and school program on the combined characteristics of training distress

The correlation coefficients between the dependent variables (i.e., the dimensions of MTDS-N) ranged between $r$ = -0.00–0.44 for males and $r$ = 0.03–0.64 for females. All

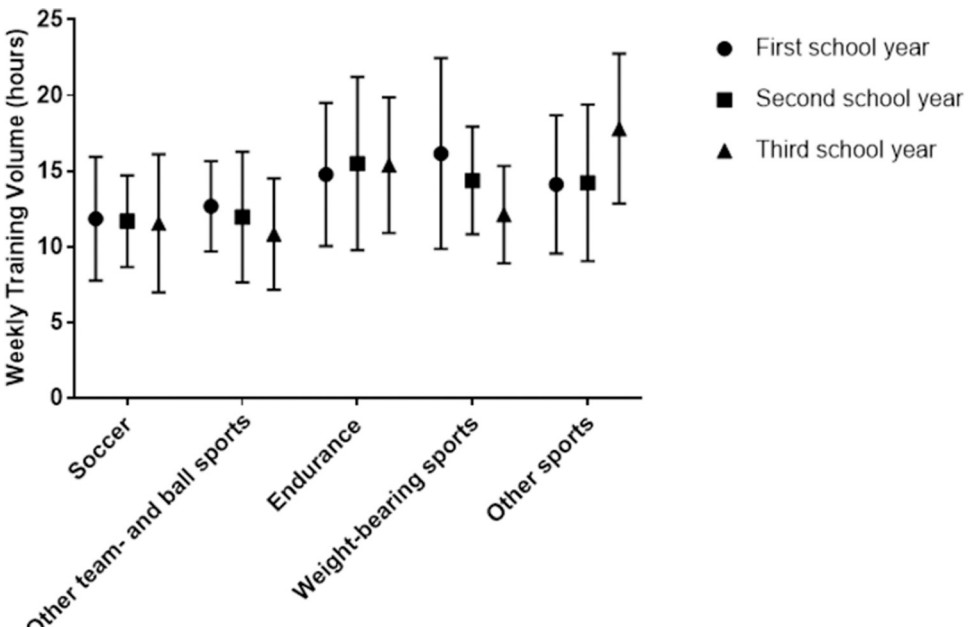

**Fig 1. Progression in weekly training volume across school years in different sport types.**

correlations were positive and significant, except for the correlation between physical symptoms and vigour for males ($r$ = -0.00) and females ($r$ = 0.03). Based on the strength of the correlations between the dimensions of MTDS-N, it was determined that it was conceptually sound

**Table 3. Mean scores for the dimensions in the Norwegian Multicomponent Training Distress Scale.**

| Dimension | Gender | *n* | *M/* MTDS-N[a] | *SD/* MTDS-N |
|---|---|---|---|---|
| Depression | Male | 308 | 1.54 | 0.64 |
| | Female | 298 | 1.76 | 0.78 |
| | Total | 606 | 1.65 | 0.72 |
| Vigour | Male | 308 | 2.60 | 0.70 |
| | Female | 298 | 2.76 | 0.74 |
| | Total | 606 | 2.68 | 0.72 |
| Physical symptoms | Male | 308 | 2.31 | 0.81 |
| | Female | 298 | 2.41 | 0.81 |
| | Total | 606 | 2.36 | 0.81 |
| Sleep disturbances | Male | 308 | 1.79 | 0.90 |
| | Female | 298 | 2.22 | 1.09 |
| | Total | 606 | 2.00 | 1.02 |
| Stress | Male | 308 | 2.45 | 0.77 |
| | Female | 298 | 2.90 | 0.84 |
| | Total | 606 | 2.67 | 0.83 |
| Fatigue | Male | 308 | 2.46 | 0.82 |
| | Female | 298 | 2.63 | 0.93 |
| | Total | 606 | 2.54 | 0.88 |

[a] The mean score of the MTDS-N, ranging between 1–5, where 1 = never/ not at all, 2 = almost never/ a little, 3 = sometimes/ moderately, 4 = fairly often/ quite a bit, and 5 = very often/extremely.

**Table 4. Results from four multivariate analyses of variance examining the effect of training volume, gender, school year, sport types, and school program.**

| MANOVA | Effect | Criteria | Value | F | Hypothesis df | Error df | p |
|---|---|---|---|---|---|---|---|
| 1 (n = 591) | TV | Λ | 0.976 | 1.18[a] | 12 | 116.00 | 0.292 |
| | Gender | Λ | 0.899 | 1.82[a] | 6 | 58.00 | 0.000** |
| | TV × Gender | Λ | 0.979 | 1.02[a] | 12 | 116.00 | 0.428 |
| 2 (n = 593) | TV | Λ | 0.977 | 1.12[a] | 12 | 1158.00 | 0.336 |
| | SY | Λ | 0.978 | 1.06[a] | 12 | 1158.00 | 0.392 |
| | TV × SY | Λ | 0.939 | 1.53 | 24 | 2021.10 | 0.048* |
| 3 (n = 593) | TV | Λ | 0.978 | 1.05[a] | 12 | 1146.00 | 0.398 |
| | ST | Λ | 0.942 | 1.43 | 24 | 200.17 | 0.082 |
| | TV × ST | Λ | 0.931 | 0.87 | 48 | 2823.46 | 0.730 |
| 4 (n = 593) | TV | Pillai's trace | 0.024 | 1.17 | 12 | 1166.00 | 0.300 |
| | Program | Pillai's trace | 0.004 | 0.40 | 6 | 582.00 | 0.877 |
| | TV × Program | Pillai's trace | 0.022 | 1.09 | 12 | 1166.00 | 0.368 |

Λ = Wilk's Lambda; TV = Weekly Training Hours; ST = Sport Types; SY = School Year.

[a] = Exact statistic.

* = $p < 0.05$.

** = $p < 0.001$

to conduct a MANOVA (S1 Table). Based on the normal Q-Q plots and considering that the MANOVA analysis is robust against the violation of normality [38], we determined that it would be safe to proceed with further analysis (S1 File).

The results from the MANOVA analyses are presented in Table 4. The first MANOVA revealed no significant multivariate effect of weekly training volume on the combined characteristics of training distress, $\lambda = 0.976$, $F (12, 1160) = 1.28$, $p = 0.292$. The multivariate effect of gender on the combined characteristics of training distress was significant irrespective of training volume per week, $\lambda = 0.899$, $F (6, 580) = 10.82$, $p < 0.001$. No significant multivariate effect across the interaction between weekly training volume and gender were observed, $\lambda = 0.979$, $F (12, 1160) = 1.02$, $p = 0.442$. Hence, only the main effect of gender was further analysed [38]. The second MANOVA indicated a significant interaction effect for weekly training volume and school year, $\lambda = 0.939$, $F (24, 2021.10) = 1.53$, $p = 0.048$. No significant effects were observed from the third or the fourth MANOVA.

**The effect of gender on perceived psychological and physiological training distress.** The assumption of homogeneity of variance-covariance was considered to be met (S2 File). The DDA results indicate that gender explained 9.5% of the variance in the composite, $\lambda = 0.905$, Chi-square (6) = 60.140, $p < 0.001$, $R^2_c = 0.095$. As shown in Table 5, stress made the most significant contribution to the equation with a standardized function coefficient of 0.86, followed by sleep disturbances and fatigue with a standardized function coefficient of 0.50 and -0.30, respectively. Physical symptoms and depression did not generate the composite outcome variable score (i.e., training distress), with standardized function coefficients of 0.00 and -0.07, respectively. Female student athletes reported higher composite variable means (i.e., training distress) (0.33 ± 1.05; CI = 0.21, 0.45) than males (-0.32 ± 0.95; CI = -0.42, -0.21). A one-way ANOVA with gender as the independent variable and the saved discriminant function scores as the dependent variable was conducted to calculate the Cohen's $d$ effect size to help quantify the magnitude of the difference [$F (1, 607) = 63.57$, $p < 0.001$, $d = 0.65$, $r = 0.31$].

**The interaction effect of weekly training volume × school year on training distress.** The assumption of homogeneity of variance-covariance was considered to be met (S2 File).

**Table 5. The contribution of each outcome variable to the linear equation.**

| Factor | Dependent variables | $R_c^2$/ % | Standardized coefficient | $r_s$ | $r_s^2$ |
|---|---|---|---|---|---|
| Gender | Depression | 0.095/ 9.5% | -0.07 | 0.47 | 0.22 |
| | Vigour | | 0.14 | 0.33 | 0.11 |
| | Physical symptoms | | 0.00 | 0.18 | 0.03 |
| | Sleep disturbances | | 0.50 | 0.67 | 0.44 |
| | Stress | | 0.86 | 0.87 | 0.76 |
| | Fatigue | | -0.30 | 0.31 | 0.09 |
| TV × SY | Depression | 0.061/ 6.1% | -0.49 | -0.22 | 0.05 |
| | Vigour | | -0.31 | -0.29 | 0.08 |
| | Physical symptoms | | 0.66 | 0.68 | 0.47 |
| | Sleep disturbances | | 0.60 | 0.49 | 0.24 |
| | Stress | | -0.20 | -0.09 | 0.01 |
| | Fatigue | | 0.14 | 0.24 | 0.06 |

$R^2_c$ = squared canonical correlation (inverse of Wilks' lambda); $r_s$ = structure coefficients; $r_s^2$ = squared structure coefficients.

The DDA results indicated the presence of a significant interaction effect of weekly training volume × school year on training distress, $\lambda = 0.939$, $F$ (24, 2021.10) = 1.53, $p = 0.048$. The interaction accounted for 6% of the variance in the composite, $R_c^2 = 0.06$. A two-way ANOVA was run to determine which groups differed on the interaction composite (S2 File). The results indicated significant differences among student athletes training 5–10 hours per week, $F$ (2, 584) = 4.393, $p = 0.013$, as well as student athletes training more than 15 hours per week, $F$ (2, 584) = 6.369, $p = 0.002$. There were no significant differences among student athletes training 10–15 hours per week. With 5–10 hours of training per week, the composite means were highest for second year student athletes (0.17 ± 1.01; CI = -0.04, 0.39) and lowest for first year student athletes (-0.31 ± 0.92; CI = -0.55, -0.07). The difference between the two groups was statistically significant ($p = 0.003$, $d = 0.48$, $r = 0.23$). For those training ≥ 15 hours per week, the composite means were highest for first year student athletes (0.33 ± 1.00; CI = 0.11, 0.56) and lowest for second year student athletes (-0.26 ± 1.18; CI = -0.49, -0.03). The difference between the two groups was statistically significant ($p < 0.001$, $d = 0.54$, $r = 0.26$). Fig 2 illustrates the interaction of weekly training volume by school year and how the training volume groups separate.

## Discussion

The primary purpose of the present investigation was to describe student athletes´ weekly training volume in Norwegian upper secondary schools and determine whether there are differences in training volume according to gender, type of sport, school program, and school year. Furthermore, we aimed to investigate whether weekly training volume, gender, type of sport, school program, or school year influence responses to the dimensions in the Norwegian Multicomponent Training Distress Scale (MTDS-N) and whether there are any interaction effects between these variables. The main findings from this study revealed no significant differences in weekly training volume for gender, school program, or school year. Nevertheless, a significant difference in weekly training volume between sport types were detected, with endurance sports having a larger training volume than more technically demanding sports. An interaction effect of weekly training volume × school year on training distress was observed where those with larger weekly training volume experienced more training distress. Further

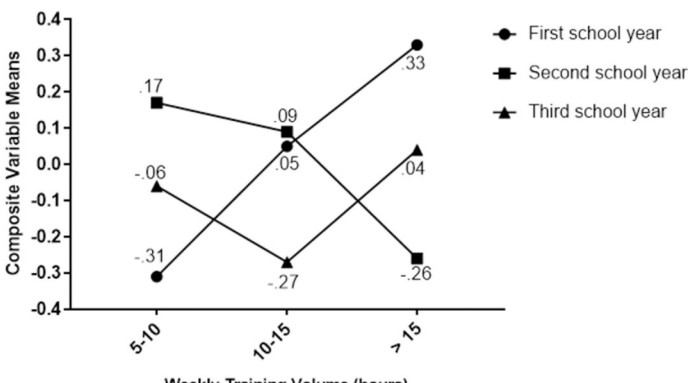

**Fig 2. Linear discriminant function plot showing the interaction of weekly training volume by school year and how the training volume groups separate.** The figure shows the means of each training volume group on the composite outcome variable that was created from the observed variables (i.e., training distress). To facilitate the interpretation of the figure, both the $r_s$ and the standardized coefficients from Table 5 could be examined.

analyses revealed a multivariate effect for gender on training distress, with females perceiving larger levels of training distress than males.

## Student athletes' weekly training volume

The significant difference in weekly training volume between the five sport types, indicated that student athletes playing soccer or other team and ball sports trained fewer hours per week than student athletes in endurance sports, weight-bearing sports, and other sports (Table 2). Previous research indicates that elite athletes in typical endurance sports train between 800–1200 hours per year [1, 19, 21, 22, 24, 25], while elite athletes in more technically demanding sports train approximately 500–700 hours per year [26–29]. As such, the findings from the present study correspond with already existing reference values for training volume. However, the reference values on training volume are for senior athletes. Interestingly, the student athletes are already close to these values at the age of 15 to 18 while combining training and school. An unexpected finding is that student athletes in weight-bearing sports have a similar weekly training volume to endurance and other sports student athletes. Based on the literature [26–29], one would expect student athletes in weight-bearing sports to train fewer hours per week, with greater similarity to those playing soccer and other team and ball sports. A possible explanation for this finding is that gymnastics was included in the weight-bearing category and is a sport requiring high training volume for high-standard performance [48].

No significant differences in weekly training volume were found between school years (Table 2). It is well documented that sustained performance development requires athletes to be exposed to a systematic increase in training load over time, while adequate recovery is also ensured [49–51]. However, as shown in Fig 1, our results indicate a significant interaction effect of sports type and school year on weekly training volume, with a decreasing trend in weekly training volume for both weight-bearing sports and other team and ball sports across school years. The trend was relatively flat in soccer, while a slight increase in endurance sports. A significant progression in training volume was observed only in the category other sports, and then only from second year to third year. Based on the trends in weekly training volume across the school years, one can question whether a long-term periodized plan was adopted to ensure progressive overload and facilitate optimal performance development [6–9]. The periods within a training macrocycle could potentially contribute to explaining this finding. It is well known that different sports have different competition periods within a training

macrocycle, which might have influenced the reported training volume. Hence, athletes in the competition season likely have less volumes with higher intensities. In comparison, athletes in the preparatory phase may have larger volumes with lower intensities where the focus is more on technical skills and the development of the general physical base [52].

## Student athletesˊ perceived psychological and physiological training distress

As shown in Table 3, scores for the different dimensions of training distress corresponded to "a little" to "moderate" amount of training distress. The results are similar to the results reported in a study of 173 student athletes competing in alpine skiing in Sweden, where the mean scores were between "a little" and "moderate" amount of training distress [53]. The Swedish student athletes' mean (± *SD*) training volume was reported to be 13.42 ± 4.07 hours per week, similar to the mean training volume in the current study of 12.76 ± 4.46 hours a week. Conversely, a study of 17 elite Australian rowers demonstrated a decline in performance in 5 km rowing combined by altered pacing strategy, suggesting an increase in fatigue. Simultaneously the total training distress scores increased significantly following four weeks of intensified training, suggesting that the athletes may have reached short-term performance decrements accompanied by psychological and physiological symptoms including mood disturbance [54]. Similar results have been found in fourteen male cyclists during a six-week training program, where increased training distress was significantly associated with increased training load (~150% of regular training load) [55]. Comparing the findings from these studies [54, 55] to the findings from this study suggest that participants training load in this study was not sufficient enough for the student athletes to reach high training distress indicated by the observed "little" to "moderate" training distress scores (Table 3). Furthermore, it has been demonstrated that those experiencing positive training adaptations are more likely to score highly on negative dimensions included in MTDS [56], which was not observed in this study. Such results suggest that the training load must be high enough to cause stress to induce the desired training adaptation. Such a concept is associated with the general adaptation syndrome (GAS) [57, 58], where adaptation is the response to stress and adequate recovery (i.e., supercompensation). This concept is also supported by the more refined stimulus-fatigue-recovery-adaptation (SFRA) theory [59, 60], which suggests that a greater stressor will result in greater fatigue and adaptation. By using MTDS-N over time, one can gather important information about athletes' psychological and physiological training distress changes and adjust their prescribed training to ensure an optimal training process. However, the authors suggest that care must be taken when interpreting psychological and physiological data. A baseline measure should always be established before decision-making, and, ideally, multiple monitoring tools should be used in parallel for a greater understanding of the athlete's overall state.

**Gender differences in perceived psychological and physiological training distress.** Irrespective of weekly training volume, the multivariate effect of gender (Table 4) indicates differences in the combined characteristics of training distress between male and female student athletes ($p < 0.001$). However, the effect size is small, accounting for approximately 10% of the variance in the composite variable. The results indicate that stress, sleep disturbances, and fatigue best discriminate between males and females (Table 5). Examining the results further indicates that depression had a relationship with the composite outcome variable, explaining 22% of its shared variance. Furthermore, the main effect observed was stress, but with a secondary contribution of sleep disturbances and depression, explaining 76%, 44%, and 22% of the shared variance, respectively. According to Main and Grove [31], depression, vigour, and stress represent measures associated with psychological overload, whereas physical symptoms,

sleep disturbances, and fatigue reflect physical and behavioural complaints related to training distress. Thus, there is a strong possibility that psychological overload could explain the difference between males and females. We acknowledge that sleep disturbances reflect physical and behavioural complaints associated with training distress. However, one can assume that a psychological overload would also contribute to sleep disturbances (e.g., difficulties falling asleep, restless sleep, and insomnia) [61].

Considering the direction of the $r_s$ in Table 5, it appears that females experienced more depression, sleep disturbances, physical symptoms, stress, fatigue, and less vigour than males. The effect size was large ($d = 0.65$, $r = 0.31$). These results corroborate findings of previous studies, which have also found female student athletes to have a higher prevalence of depressive symptoms [62] and greater fatigue levels with lower vigour levels [63, 64] compared to male student athletes. Furthermore, female student athletes have also been found to have relatively higher psychological distress levels [65]. Studies have also indicated that sleep disturbances are more prevalent in adolescent females [66, 67], with gender differences emerging after menses onset [68]. In addition, sleep disturbances among female athletes are more prevalent than for male athletes [69]. These findings, including the results from the current study, can be explained by maturation and growth differences between the two genders. Due to the increase of estrogen production and a slower rate of muscles development, girl adolescents may find it more challenging to adapt to the somatic growth spurt in the context of their sport or physical activity [70]. For example, the increase of estrogen production leads to increases in body fat deposition, breast development, and widening of the hips, which further contribute to changes in female body shape, the center of gravity, and strength-to-body mass ratio, which may negatively affect sports performance [71]. Conversely, males typically experience physical performance improvements during adolescence. The marked increase in hormonal concentrations in boys (i.e., testosterone, growth hormone, and insulin-like growth factor) typically leads to a significant increase in muscle mass and longer bones (i.e., widening of the shoulders and longer appendicular skeleton bones), leading to an acceleration in strength gains [72]. In addition, these developments in boys and girls increase the demand from the circulatory and respiratory systems to supply oxygen to skeletal muscle mitochondria for energy production. This causes an increase in cardiac output (i.e., increased blood volume, myocardial contractility, ventricular compliance, and angiogenesis), which, in turn, contribute to increases in peak oxygen uptake [73].

Furthermore, puberty can be psychologically challenging, especially for females [13]. At 15 years of age, a strong association has previously been found between menarche and mental distress [74]. However, this association was no longer statistically significant three years later among the same girls. Student athletes start upper secondary school the year they turn 16, indicating that extra consideration may be needed for females in their first year of upper secondary school. The effect of being different might be more noticeable during puberty with rapid body changes, compared to later stages when body dissatisfaction may be more related to elevated adiposity and living in an environment where the ideal is to be thin [75]. However, additional research is needed to test different variables that explain potential gender differences and mental health relationships in sports [76]. It should be noted that the polarity in willingness to report any psychological symptoms is a familiar issue when comparing psychological distress levels between genders [65]. Regardless, the available findings confirm the need for increased attention from those involved with female student athletes (e.g., parents, teachers, and club coaches) in order to prevent negative training and health outcomes.

Irrespective of gender differences, it is essential to emphasize that the student athletes´ self-reported training distress was generally low to moderate in the current study. Table 3 shows that the overall mean score was 2.18 and 2.45 for males and females, respectively,

corresponding to "a little" to "a moderate" amount of training distress. In addition, a systematic review and meta-analysis found that symptoms of anxiety and depression were significantly lower among adolescents involved in sport than those who did not participate in sport, although the effect size was small [76]. Interestingly, of the six dimensions included in MTDS-N, depression had the lowest mean score of 1.54 and 1.76 for both males and females, respectively. Other studies have found that the prevalence of psychological distress among young elite athletes is lower than for general population controls [77]. Hence, elite sport participation does not appear to be related to elevated psychological distress levels [78]. Davis et al. [53] also concluded that student athletes´ stress levels were relatively low, which does not support the traditional assumption in sport psychology that student athletes combining both school and sports are more vulnerable to increased stress levels [79].

## The interaction effect of weekly training volume × school year on perceived psychological and physiological training distress

The interaction between weekly training volume and school year ($p = 0.048$) indicates a difference in perceived training distress between school years with different training volumes per week (Table 4). In other words, one factor influences the effects of the other factor at a particular level [80]. Nevertheless, the interaction's effect size was small (Table 5), accounting for only 6% of the variance in the composite variable (i.e., training distress). Furthermore, the observed interaction effect was mainly for physical symptoms but with a secondary contribution of sleep disturbances, explaining 47% and 24% of the shared variance, respectively. Hence, the difference is mainly explained by physical and behavioural complaints associated with training distress [31]. This finding is contrary to the effect of gender, where the difference was explained primarily by psychological overload.

As shown in Fig 2, first year student athletes had significantly ($p = 0.003$, $d = 0.48$, $r = 0.23$) lower perceived training distress than second year student athletes with 5–10 weekly training hours. Conversely, amongst those training $\geq 15$ hours per week, first year student athletes had significantly ($p < 0.001$, $d = 0.54$, $r = 0.26$) higher perceived training distress compared to student athletes in second year. In other words, the larger the training volume, the greater the perceived training distress among first year student athletes. This finding can be explained by two different hypotheses. Firstly, student athletes may adapt to the training load, so that by their second year they experience less training distress than in their first year, despite similar training volumes ($\geq 15$ hours). In light of the GAS concept [57, 58] and SFRA theory [59, 60], student athletes likely experience an adaptation during the transition from first year to second year. However, comparing the results between second-and third-year student athletes indicates that this adaptation does not continue after the second year. This could be due to a lack of change in training intensity, since we know that training volume was the same across the school years in the different sport types. It is well documented that one must influence either training volume and/or training intensity in order to improve performance [2]. The second hypothesis is that student athletes experience a higher level of training distress in their first year because they were not prepared for the increased training load they encounter when transitioning from lower secondary school to upper secondary school [14–16]. The weekly training volume may not have been appropriately adjusted to student athletes in their first year; hence, there is a possibility that the training load was too high, explaining the increased levels of training distress. Hence, practitioners (i.e., club coaches and school coaches) should carefully monitor and manage athletes' stress and recovery to avoid harmful outcomes. Further, to prepare student athletes for the increased training load they encounter when they are enrolled into upper secondary school, practitioners should cooperate and design an individualized training

plan ensuring an appropriate progression in training load. Such a plan would also help to maintain performance development throughout second and third year. With low to moderate levels of training distress, as shown in Table 3, there may be room to increase the training intensity across the school years. By regularly monitoring student athletes, coaches can evaluate how they are coping with and tolerating the training load and make necessary adjustments to optimize performance capacity [7, 18].

## Strengths, limitations and future research directions

The strength of the present study is the large number of participants from different counties in Norway. Further, DDA was conducted as a multivariate post-hoc analysis for evaluating the MANOVA effects, which has been recommended when running several ANOVAs to test mean differences [40, 44, 45]. However, some limitations need to be considered; first, the present study involved a self-reported questionnaire and, as such, response bias may have influenced the results. Second, weekly training volume was also self-reported and may be somewhat inaccurate, and the type of exercises and training intensities were not registered. Third, no similar studies have previously been conducted in an equivalent population, making it hard to compare the present results. However, this study can be seen as a starting point in establishing a norm for this population. Hence, future research should use a longitudinal design with student athletes reporting daily training and weekly perceived training distress with the MTDS questionnaire. Doing so makes it possible to detect spikes in perceived training distress and improve training periodisation. Finally, future research should also focus on other factors explaining performance development in student athletes, such as general life load and the prevalence of injury and health problems.

## Conclusions and practical implications

To our knowledge, this is the first study to describe weekly training volume and perceived psychological and physiological training distress in student athletes enrolled in the subject "top-standard sport" in Norwegian upper secondary schools. Research to date, including the current study results, suggests the need for increased attention from practitioners involved with female student athletes to prevent adverse health outcomes and decreased performance. Practitioners should adhere to a conceptual framework for the periodization of training in order to facilitate a progressive training stimulus leading to positive adaptation and performance development. A long-term training plan is essential to smooth the transition from lower secondary school to upper secondary school and ensure that the training load is appropriately adjusted to match each individual´s anthropometric, physical, and metabolic characteristics. Regular monitoring with a user-friendly questionnaire such as MTDS-N can help practitioners preserve student athletes' physiological and psychological well-being and ensure positive performance development.

## Supporting information

**S1 Table. Pearson bivariate correlations among study variables.**
(DOCX)

**S2 Table. The categorization of the different sports in the present study.**
(DOCX)

**S1 File. Testing normality.**
(DOCX)

**S2 File. Descriptive discriminant analysis.**
(DOCX)

**S1 Dataset. Source data.**
(XLSX)

## Author Contributions

**Conceptualization:** Cathrine Nyhus Hagum, Espen Tønnessen, Shaher A. I. Shalfawi.

**Data curation:** Cathrine Nyhus Hagum.

**Formal analysis:** Cathrine Nyhus Hagum.

**Funding acquisition:** Shaher A. I. Shalfawi.

**Investigation:** Cathrine Nyhus Hagum, Espen Tønnessen, Shaher A. I. Shalfawi.

**Methodology:** Cathrine Nyhus Hagum, Espen Tønnessen, Shaher A. I. Shalfawi.

**Project administration:** Cathrine Nyhus Hagum, Shaher A. I. Shalfawi.

**Supervision:** Espen Tønnessen, Shaher A. I. Shalfawi.

**Validation:** Cathrine Nyhus Hagum, Espen Tønnessen, Shaher A. I. Shalfawi.

**Visualization:** Cathrine Nyhus Hagum, Espen Tønnessen, Shaher A. I. Shalfawi.

**Writing – original draft:** Cathrine Nyhus Hagum.

**Writing – review & editing:** Cathrine Nyhus Hagum, Espen Tønnessen, Shaher A. I. Shalfawi.

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
