## [Decision Letter · Decision Letter 0]

31 Aug 2021

PONE-D-21-13902

Progression in training volume and perceived psychological and physiological training distress in Norwegian student athletes

PLOS ONE

Dear Cathrine Nyhus Hagum,

Thank you for submitting your manuscript to PLOS ONE. After careful consideration, we feel that it has merit but does not fully meet PLOS ONE’s publication criteria as it currently stands. Therefore, we invite you to submit a revised version of the manuscript that addresses the points raised during the review process.

We recommend that you pay particular attention to reviewer 2's comments. For example, as highlighted, important methodological indications are missing regarding (for example): the measure used to assess training volume, criterions that influenced the exclusion of participants, how cutpoints for hours/week for training volume groups were determined, etc.

Please submit your revised manuscript by September 17th, 2021. If you need more time than this to complete your revisions, please reply to this message or contact the journal office at plosone@plos.org. Please include the following items when submitting your revised manuscript:

We look forward to receiving your revised manuscript.

Kind regards,

Jonathan Smith, Ph.D.

Academic Editor

PLOS ONE

1. Please ensure that your manuscript meets PLOS ONE's style requirements, including those for file naming. The PLOS ONE style templates can be found at https://journals.plos.org/plosone/s/file?id=wjVg/PLOSOne_formatting_sample_main_body.pdf and https://journals.plos.org/plosone/s/file?id=ba62/PLOSOne_formatting_sample_title_authors_affiliations.pdf.

2. We noted in your submission details that a portion of your manuscript may have been presented or published elsewhere. (Hagum and Shalfawi, 2020) Please clarify whether this [conference proceeding or publication] was peer-reviewed and formally published. If this work was previously peer-reviewed and published, in the cover letter please provide the reason that this work does not constitute dual publication and should be included in the current manuscript.

Reviewers' comments:

Reviewer's Responses to Questions

**Comments to the Author**

1. Is the manuscript technically sound, and do the data support the conclusions?

Reviewer #1: Yes

Reviewer #2: Yes

Reviewer #3: Yes

2. Has the statistical analysis been performed appropriately and rigorously? 

Reviewer #1: Yes

Reviewer #2: Yes

Reviewer #3: Yes

3. Have the authors made all data underlying the findings in their manuscript fully available?

Reviewer #1: Yes

Reviewer #2: Yes

Reviewer #3: Yes

4. Is the manuscript presented in an intelligible fashion and written in standard English?

Reviewer #1: Yes

Reviewer #2: Yes

Reviewer #3: Yes

5. Review Comments to the Author

Reviewer #1: The authors of this study used a cross-sectional survey to examine training volume and training distress in a large sample of Norwegian student athletes. Weekly training volume differed by sport, but not gender, school program, or school level. Gender explained 9.5% of the variance in combined physical and psychological distress. The authors conclude that regular monitoring with an instrument such as the Multicomponent Training Distress Scale can help coaches adjust training load as needed to optimize performance and physical/psychological wellbeing.

This manuscript was clear, concise, and easy to read, and the methods were appropriate for the purposes of the study. Furthermore, the authors provided practical implications for coaches and sport administrators based on their results and other literature. I have a few suggestions to further strengthen this paper.

Methods:

1. Please include a little more detail on the procedure so that readers don’t have to look up the other paper. At a minimum, I’d like to know at what point in the school year data collection took place, and where/how the questionnaires were completed (e.g., during class, at sport practice, online, etc.)

2. The categories of soccer, other team- and ball sports, and endurance sports make sense, but I’m not convinced by the separation of sports into “weight-bearing” and “other.” Surely gymnastics and cheerleading (and perhaps figure skating and diving) share more similarities than gymnastics and track and field? And shouldn’t snowboarding go in the same category as alpine skiing?

Results:

1. Please give effect sizes (e.g., partial eta squared) for the ANOVAs and MANOVAs.

2. It would be nice to have meaningful/significant mean differences indicated in the tables and figures in some way.

Discussion:

1. Line 276 – Add an apostrophe after athletes.

2. Lines 291-295 – The authors describe the training volume of the athletes in the weight-bearing category as unexpected, but I wonder if this is due to the inclusion of gymnastics in this category. Gymnastics is known for being an early specialization sport that requires high volumes of training from a fairly young age.

3. Gender Differences – I’ve read that adult women generally have poorer sleep quality and more sleep disturbances compared to men. This might be worth mentioning, particularly if this gender difference also applies in adolescents and/or athletes.

4. Page 22 – Although limitations are clearly stated and practical implications are provided, there isn’t much in the way of suggestions for future research. Please discuss more explicitly some potential avenues for future investigations.

Reviewer #2: This is an interesting manuscript aimed to describe weekly training volume in Norwegian upper secondary schools student athletes according to gender, type of sport, school program, and school level. In addition this manuscript aimed to investigate whether weekly training volume, gender, type of sport, school program, or school level influence perceived training distress.

I consider that the subject addressed in this paper is interesting and relevant to improve periodization of training in young population. Although the design and the analysis developed are appropriate, several major issues about the manuscript should be addressed to improve writing quality and to fulfill STROBE (Strengthening the reporting of Observational studies in Epidemiology) guidelines for cross-sectional studies.

Various sections should be improved to guarantee a better interpretation of the information. Specific comments are presenting next.

To fulfill STROBE (Strengthening the reporting of Observational studies in Epidemiology) guidelines, it should be indicated the study’s design with a commonly used term in the title or the abstract. In addition to the objectives, it should be stated any pre-specified hypotheses in the Introduction section. In Methods section the key elements of study design should be presented, including relevant dates and periods of recruitment. It should be explained the sample size calculation and how the study size was arrived at. In results section, sociodemographic characteristics of the sample should be explained. In addition to the study limitations, the strengths of the study should be indicated. Maybe the sections conclusions and implications for practice could be separated to properly understand the specific information of both sections.

Abtract:

Lines 20 to 21: Authors only presented the instrument used to evaluate perceived training distress, but did not present the instrument used to evaluate training volume, which is also an important variable of the study. Please indicate how training volume was evaluated in the abstract.

Line 22: (M age = 17.29, SD = .94). To maintain the style used in results section, please presented this result as M ± SD.

Line 24: “No significant differences were found for gender, school level, or school program”. I can interpret that the variable authors are analyzing is training volume, but it must be indicated in the text.

Introduction:

Authors should summarize the information to highlight the main arguments for avoiding that secondary information get reader distracted from the main argument.

Methods:

The design of this study should be specifically explained, including relevant dates of data collection and periods of recruitment. Maybe it would be appropriated including a subsection called “Study design” previously to “Participants” subsection.

Lines 92 to 94: it should be stated the reasons why potential participants with ≤4 hours and ≥30 hours of training per week were excluded. The first criterion seems obvious for guarantee a minimum training volume, but why authors established a maximum training volume. Is this decision based in scientific evidence? If the answer is yes, please provide a reference.

Lines 102 to 105: “Data regarding age, gender, country, school name, study program, school level, type of sport and weekly training volume were collected in addition to the questionnaire. The instrument and data collection procedure are fully described in (35)”. The MTDS questionnaire was fully described in this reference as the authors commented, but the questions regarding type of sport and weekly training volume must be explained in order to properly interpret the data obtained for these variables.

Lines 108 to 109. Please indicate that the factor vigor is a factor from the MTDS questionnaire and requires reversing it.

Lines 123 to 124: Authors should explain the reasons why they establish a cutpoint of 5, 10 and 15 hours/week for training volume groups. Is this decision based in scientific evidence? If the answer is yes, please provide a reference.

Results:

Results section should start showing the main descriptive characteristics of the sample, i.e.: % of students in each type of sport, school program, school level and another descriptive characteristics with relevance to the study aim.

Lines 161 to 163: Authors should indicate if when applied Bonferroni adjustment it was found or not significant differences in any comparisons between subgroups.

Discussion:

When starting each paragraph authors have to interpret the results obtained instead of mainly present them to avoid repeat information from Results section.

Lines 266 to 271. Instead of repeat the purposes of the study authors should start the discussion section highlighting the main findings of the study based on the two purposes of it.

Line 275. A short general statement about the practical implication of this highlight could improve this first paragraph of Discussion section.

Line 291 to 295. Could authors establish an interpretation of this difference and provide a possible explanation of it?

Lines 299 to 304. Weight-bearing sport and other team and ball sport have a decreasing trend and soccer has a relative flat trend, while other sports have a progression in training volume. What could explain it? Perhaps the interest of coaches in likening the training load to adult or elite players could influence it? Could authors go in depth in this fact?

Lines 314 to 315. To maintain the style used in results section, please presented this result as M ± SD.

Conclusion

Authors should specify the main findings highlighted by this study. Maybe the sections implications for practice and conclusions could be separated to properly understand the specific information of both sections.

Reviewer #3: A great study, nicely executed and well written. Some minor formatting issues with the final column of Table 2.

6. PLOS authors have the option to publish the peer review history of their article (what does this mean?). If published, this will include your full peer review and any attached files.

Reviewer #1: No

Reviewer #2: No

Reviewer #3: **Yes: **Luana Main

---

## [Author Response · Author response to Decision Letter 0]

17 Sep 2021

Response to Reviewers

Dear Dr. Jonathan Smith,

Thank you for the opportunity to submit a revised version of the manuscript titled Progression in training volume and perceived psychological and physiological training distress in Norwegian student athletes to PLOS ONE. We appreciate the time and effort you and the reviewers have dedicated to providing valuable feedback on the submitted manuscript. We have addressed the raised concerns and hope our responses and revisions will meet your expectations. We have highlighted the changes within the manuscript.

Below is a point-by-point response to the academic editor and the reviewers' comments and concerns. All changes in the manuscript are marked up with red colour.

Journal requirements

1. Please ensure that your manuscript meets PLOS ONE's style requirements, including those for file naming. The PLOS ONE style templates can be found at https://journals.plos.org/plosone/s/file?id=wjVg/PLOSOne_formatting_sample_main_body.pdf and https://journals.plos.org/plosone/s/file?id=ba62/PLOSOne_formatting_sample_title_authors_affiliations.pdf.

Response to the academic editor:

We have double-checked the manuscript to meet PLOS ONE's style requirements. 

2. We noted in your submission details that a portion of your manuscript may have been presented or published elsewhere. (Hagum and Shalfawi, 2020) Please clarify whether this [conference proceeding or publication] was peer-reviewed and formally published. If this work was previously peer-reviewed and published, in the cover letter please provide the reason that this work does not constitute dual publication and should be included in the current manuscript.

Response to the academic editor:

In the submitted research article, we have used the same data as in Hagum and Shalfawi (2020), which is peer-reviewed and published. However, Hagum and Shalfawi (2020) is a methodological article investigating the factorial validity of the MTDS-N questionnaire. The current research article raises a new set of research questions which was part of the pre-approved protocol by the Norwegian Social Science Data Services (NSD) (Project number: 836079) and the Regional Committees for Medical and Health Research Ethics (REK) (project number: 54584) prior to data collection. Hence, this work does not constitute dual publication.

Reviewer #1

Review Comments to the Author

This manuscript was clear, concise, and easy to read, and the methods were appropriate for the purposes of the study. Furthermore, the authors provided practical implications for coaches and sport administrators based on their results and other literature. I have a few suggestions to further strengthen this paper.

Response to the reviewer:

Thank you for taking the time to assess the manuscript. We have addressed the raised concerns and hope our responses and revisions will meet your expectations.

Methods:

1. Please include a little more detail on the procedure so that readers don't have to look up the other paper. At a minimum, I'd like to know at what point in the school year data collection took place, and where/how the questionnaires were completed (e.g., during class, at sport practice, online, etc.)

Response to the reviewer:

We thank the reviewer for pointing this out. We have updated the manuscript with the following text: All upper secondary schools that offer the optional program subject top-level sport in Norway (n = 119) were invited to participate in the present study. The MTDS-N was distributed electronically using SurveyXact version 8.0 (Ramboll, n.d.) to the school management who agreed to participate (n = 34, 28.6%). Further, the school management distributed the questionnaire electronically to the student-athletes at their respective schools (n = 23, 19.3%). The data collection took place during class and started in March 2020 and ended in May 2020. To assess training volume, student athletes reported their current weekly training hours. In addition, the survey included questions regarding age, gender, county, school name, study program, school level, and main type of sport.

2. The categories of soccer, other team- and ball sports, and endurance sports make sense, but I'm not convinced by the separation of sports into "weight-bearing" and "other." Surely gymnastics and cheerleading (and perhaps figure skating and diving) share more similarities than gymnastics and track and field? And shouldn't snowboarding go in the same category as alpine skiing?

Response to the reviewer:

This is an important and reasonable argument. It was challenging to place some of the sports in the present study. The different sports included in the "weight-bearing" category had the most common features concerning the type of load dynamics, muscle contraction, and duration. Performance in these sports requires maximum power of both the anaerobic alactic system and the anaerobic lactic system. These were the criteria that we considered when placing them in this category. We could have done it in other ways, but this was the choice we made.

We considered placing snowboard together with alpine skiing. However, the choice fell on "other" because of the number of participants reporting snowboarding as their main sport and since snowboarding involves more acrobatic elements than requirements for strength and load dynamics as in the other sports placed in the "weight-bearing" category.

A reason for creating a category with "other" sports was that several sports had a low n. For example, all sports in the category "other" have an n ranging between 0.2-1.9% of the total sample size of 608 participants. To do sensible statistical analyses, we had to ensure enough participants in each sport category.

S2 Table, "The categorization of the different sports in the present study," is updated with track changes. In this table, we had forgotten to implement five different sports (jetski, dance, motocross, climbing, and figure skating) in the "other sports" category. 

Results:

1. Please give effect sizes (e.g., partial eta squared) for the ANOVAs and MANOVAs.

Response to the reviewer:

We agree and have presented effect sizes by calculating Cohen's d for the ANOVAs (please see lines 212 to 218) and the two-way ANOVA (please see lines 239 to 245). Further, effect sizes for the MANOVAs are presented with squared canonical correlation (R2c), which is inverse of Wilks' lambda. This variance-accounted-for effect size is analogous to R2 in multiple regression or eta squared (η2) in ANOVA (Sherry, 2006).

2. It would be nice to have meaningful/significant mean differences indicated in the tables and figures in some way.

Response to the reviewer:

We have made sure that the meaningful/significant mean differences are explicitly stated in the manuscript text. Due to the journal requirements, the results could not be double presented. We are hoping to meet your expectations.

Discussion:

1. Line 276 – Add an apostrophe after athletes.

Response to the reviewer:

We have made the change, and an apostrophe after athletes in line 336 is added in the manuscript. 

2. Lines 291-295 – The authors describe the training volume of the athletes in the weight-bearing category as unexpected, but I wonder if this is due to the inclusion of gymnastics in this category. Gymnastics is known for being an early specialization sport that requires high volumes of training from a fairly young age.

Response to the reviewer:

We thank the reviewer for this valuable interpretation. We agree and have updated the manuscript: A possible explanation for this finding is that gymnastics was included in the weight-bearing category and is a sport requiring high training volume for high-level performance (Caine & Harringe, 2013).

3. Gender Differences – I've read that adult women generally have poorer sleep quality and more sleep disturbances compared to men. This might be worth mentioning, particularly if this gender difference also applies in adolescents and/or athletes.

Response to the reviewer:

This observation is correct. We have included the following text in the section discussing gender differences: Studies have also indicated that sleep disturbances are more prevalent in adolescent females (Galland et al., 2017; Hysing et al., 2013), with gender differences emerging after menses onset (Johnson et al., 2006). The female gender has also been identified as a risk factor for sleep disturbances in athletes (Walsh et al., 2021).

4. Page 22 – Although limitations are clearly stated and practical implications are provided, there isn't much in the way of suggestions for future research. Please discuss more explicitly some potential avenues for future investigations.

Response to the reviewer:

We agree and have updated the section "Limitations" with potential avenues for future research: Future research should use a longitudinal design with student athletes reporting daily training and weekly perceived training distress with the MTDS questionnaire. This could make it possible to detect spikes in perceived training distress and improve the periodization of training in student athletes. Future research should also focus on other factors explaining performance development, such as general life load and the prevalence of injury and health problems. 

 

Reviewer #2

Review Comments to the Author 

I consider that the subject addressed in this paper is interesting and relevant to improve periodization of training in young population. Although the design and the analysis developed are appropriate, several major issues about the manuscript should be addressed to improve writing quality and to fulfill STROBE (Strengthening the reporting of Observational studies in Epidemiology) guidelines for cross-sectional studies.

Various sections should be improved to guarantee a better interpretation of the information. Specific comments are presenting next.

Response to the reviewer:

Thank you for taking the time to assess the manuscript. We have addressed the raised concerns and hope our responses and revisions will meet your expectations.

1. To fulfill STROBE (Strengthening the reporting of Observational studies in Epidemiology) guidelines, it should be indicated the study's design with a commonly used term in the title or the abstract. 

Response to the reviewer:

The title is adjusted, and the study's design is now included in the title: "Progression in training volume and perceived psychological and physiological training distress in Norwegian student athletes: a cross-sectional study." The study's design is also included in the abstract (please see line 32). 

2. In addition to the objectives, it should be stated any pre-specified hypotheses in the Introduction section.

Response to the reviewer: 

We thank the reviewer for pointing this out. We agree and have updated the manuscript. Please see lines 105 to 116. 

In the section "Conclusion," we have answered the different hypotheses.

3. In Methods section the key elements of study design should be presented, including relevant dates and periods of recruitment. 

Response to the reviewer:

We agree and have updated the manuscript. Relevant dates and periods of recruitment are included in the manuscript. Please see lines 149 to 158. 

4. It should be explained the sample size calculation and how the study size was arrived at. 

Response to the reviewer:

We have revised the manuscript and implemented a paragraph explaining the sample size calculation (lines 119-132): The point of stability approach was used to estimate the sample size (Hirschfeld et al., 2014; Kretzschmar & Gignac, 2019; Schönbrodt & Perugini, 2013). This approach ensures that the deviation between the estimated sample and the population parameter is stable (small) and is predicted to remain small at a stable statistical power (80%) (Kretzschmar & Gignac, 2019; Schönbrodt & Perugini, 2013). According to Cohen (Cohen, 1988), to ensure a small stability, the corridor of stability should not exceed a small correlation of 0.10. Schönbrodt and Perugini (Schönbrodt & Perugini, 2013) suggested that the minimum number needed to reach the point of stability would be 240–250 participants. According to Kretzschmar and Gignac (Kretzschmar & Gignac, 2019) the point-estimates of the correlation was stabilized at a sample size of 220 with perfect reliability (omega, ω = 1.0) of both latent factors and a population correlation of p = 0.20. Because perfect reliability is almost never attained, the authors proposed that the required sample at a population correlation of p = 0.20 and reliability of ω = 0.7 would be ≥ 490 participants (Kretzschmar & Gignac, 2019). Hirschfeld, Brachel and Thielsch have reported similar results with the recommended sample size to reach a point of stability was > 500 participants (Hirschfeld et al., 2014). Consequently, the sample size that was required in this study was to be more or equal to the recommendations from comparable studies (i.e., n ≥ 500).

5. In results section, sociodemographic characteristics of the sample should be explained. 

Response to the reviewer:

We have included a table (Table 1) that explains the sociodemographic characteristics of the sample. 

6. In addition to the study limitations, the strengths of the study should be indicated. 

Response to the reviewer:

We have adjusted the section and implemented the strengths of the study. 

7. Maybe the sections conclusions and implications for practice could be separated to properly understand the specific information of both sections.

Response to the reviewer:

We agree with the reviewer and have separated the paragraph.

Abstract

1. Lines 20 to 21: Authors only presented the instrument used to evaluate perceived training distress, but did not present the instrument used to evaluate training volume, which is also an important variable of the study. Please indicate how training volume was evaluated in the abstract.

Response to the reviewer:

We have updated the abstract (please see line 32) and the "instrument and procedure" section, explaining how we collected information about the student athletes training volume: To assess training volume, student athletes reported their current weekly training hours. In addition, the survey included questions regarding age, gender, county, school name, study program, school level, and main type of sport.

2. Line 22: (M age = 17.29, SD = .94). To maintain the style used in results section, please presented this result as M ± SD.

Response to the reviewer:

The manuscript is adjusted, and the result is presented as M ± SD.

3. Line 24: "No significant differences were found for gender, school level, or school program". I can interpret that the variable authors are analyzing is training volume, but it must be indicated in the text.

Response to the reviewer:

The manuscript is adjusted, and training volume is indicated in the text. 

Introduction:

1. Authors should summarize the information to highlight the main arguments for avoiding that secondary information get reader distracted from the main argument.

Response to the reviewer:

A major part of the introduction is rewritten, maintaining the purpose of the study in mind. 

Methods:

1. The design of this study should be specifically explained, including relevant dates of data collection and periods of recruitment. Maybe it would be appropriated including a subsection called "Study design" previously to "Participants" subsection.

Response to the reviewer:

The design of the study is presented in the title and the abstract (please see line 32). Further, we have included relevant data collection dates and recruitment periods: All upper secondary schools that offer the optional program subject top-level sport in Norway (n = 119) were invited to participate in the present study. The MTDS-N was distributed electronically using SurveyXact version 8.0 (Ramboll, n.d.) to the school management who agreed to participate (n = 34, 28.6%). Further, the school management distributed the questionnaire electronically to the student-athletes at their respective schools (n = 23, 19.3%). The data collection took place during class and started in March 2020 and ended in May 2020. 

2. Lines 92 to 94: it should be stated the reasons why potential participants with ≤4 hours and ≥30 hours of training per week were excluded. The first criterion seems obvious for guarantee a minimum training volume, but why authors established a maximum training volume. Is this decision based in scientific evidence? If the answer is yes, please provide a reference.

Response to the reviewer:

In the preliminary analyses, SPSS marked two participants (with 40 and 37.5 hours of training a week) as outliners. One participant with 30 hours of training a week was marked as an extreme value. This participant reported soccer as the main sport. Soccer players of this age usually have 10-15 hours of training per week. This would result in self-training up to 15-20 hours a week, which we find unlikely. Hence, we considered this as a fictitious value. The decision might have been different if the reported sport was, for example, gymnastics or swimming. We decided to exclude outliers and the participant with 30 hours of training to obtain sound analyses. Weekly training volume was self-reported and is a limitation of this study. The supplementary file 1.2 testing normality indicates no outliers after excluding these three outliers. We have adjusted the manuscript, describing why we excluded the participants with ≤4 hours and ≥30 hours of training per week.

3. Lines 102 to 105: "Data regarding age, gender, country, school name, study program, school level, type of sport and weekly training volume were collected in addition to the questionnaire. The instrument and data collection procedure are fully described in (35)". The MTDS questionnaire was fully described in this reference as the authors commented, but the questions regarding type of sport and weekly training volume must be explained in order to properly interpret the data obtained for these variables.

Response to the reviewer:

We have updated the manuscript, explaining how we collected information about the student athletes type of sport and weekly training volume: To assess training volume, student athletes reported their current weekly training hours. In addition, the survey included questions regarding age, gender, county, school name, study program, school level, and main type of sport. 

4. Lines 108 to 109. Please indicate that the factor vigor is a factor from the MTDS questionnaire and requires reversing it.

Response to the reviewer:

We have changed the sentence to: First, the factor vigor from the MTDS questionnaire, with positive scores, was reversed.

5. Lines 123 to 124: Authors should explain the reasons why they establish a cutpoint of 5, 10 and 15 hours/week for training volume groups. Is this decision based in scientific evidence? If the answer is yes, please provide a reference.

Response to the reviewer:

This is an interesting comment from the reviewer. The cutpoint of 5, 10, and 15 hours of training per week was done to ensure relatively equal group sizes (Finch & French, 2013). Further, an equal progression in training load would provide logic in the interpretation of data. The manuscript is revised, and we have included the reason for the classification of training hours. Please see lines 178 to 180.

Results:

1. Results section should start showing the main descriptive characteristics of the sample, i.e.: % of students in each type of sport, school program, school level and another descriptive characteristics with relevance to the study aim.

Response to the reviewer:

We have included a table (Table 1) showing the main characteristics of the sample in the present study. 

2. Lines 161 to 163: Authors should indicate if when applied Bonferroni adjustment it was found or not significant differences in any comparisons between subgroups.

Response to the reviewer:

We have adjusted the manuscript and stated that Bonferroni adjustment was applied when subgroups were compared.

Discussion:

1. When starting each paragraph authors have to interpret the results obtained instead of mainly present them to avoid repeat information from Results section.

Response to the reviewer:

We have revised the discussion by removing repeated results from the result chapter and maintain the information necessary for the discussion.

2. Lines 266 to 271. Instead of repeat the purposes of the study authors should start the discussion section highlighting the main findings of the study based on the two purposes of it.

Response to the reviewer:

We have removed the purposes of the study and start the discussion section by highlighting the main findings of the study. 

3. Line 275. A short general statement about the practical implication of this highlight could improve this first paragraph of Discussion section.

Response to the reviewer:

We agree with the reviewer and have included a short general statement about the practical implication. 

4. Line 291 to 295. Could authors establish an interpretation of this difference and provide a possible explanation of it?

Response to the reviewer:

A possible explanation for this finding can be that gymnastics is included in the weight-bearing category. Gymnastics is known for being an early specialization sport, in which young children are exposed to a high sports-specific training volume (Moeskops et al., 2019).

5. Lines 299 to 304. Weight-bearing sport and other team and ball sport have a decreasing trend and soccer has a relative flat trend, while other sports have a progression in training volume. What could explain it? Perhaps the interest of coaches in likening the training load to adult or elite players could influence it? Could authors go in depth in this fact?

Response to the reviewer:

We thank the reviewer for the input and encouragement to reflect on this finding. One possible explanation can be linked to the type of competition in the given sports. For example, soccer has a more stable training load during the year than the sports in the "other" category, which is more preparatory. 

In endurance and other individual sports, a more traditional periodization is often used where one has a progression in the number of hours of training per week (Myakinchenko et al., 2020; Treff et al., 2017; Tønnessen et al., 2014). We can assume that team and ball sports have not used periodization models to the same degree because the competition season is long, and the preparation period is short. Team and ball sports often have an equal training time from week to week, while other sports are not as tied to fixed training times, which also could explain the finding.

Another explanation might be that coaches in the same sport have different philosophies regarding training load. Furthermore, the time of the data collection could explain the finding from the present study. 

6. Lines 314 to 315. To maintain the style used in results section, please presented this result as M ± SD.

Response to the reviewer:

We have adjusted the manuscript, and the result is presented as M ± SD.

Conclusion:

1. Authors should specify the main findings highlighted by this study. Maybe the sections implications for practice and conclusions could be separated to properly understand the specific information of both sections.

Response to the reviewer:

We agree with the reviewer and have revised the manuscript. The main finding is highlighted. Further, we have separated the paragraph.

 

Reviewer #3

Review Comments to the Author 

A great study, nicely executed and well written. Some minor formatting issues with the final column of Table 2.

Response to the reviewer:

Thank you very much.

---

## [Decision Letter · Decision Letter 1]

9 Dec 2021

PONE-D-21-13902R1Progression in training volume and perceived psychological and physiological training distress in Norwegian student athletes: a cross-sectional studyPLOS ONE

Dear Dr. Nyhus Hagum,

Thank you for submitting your manuscript to PLOS ONE. After careful consideration, we feel that it has merit but does not fully meet PLOS ONE’s publication criteria as it currently stands. Therefore, we invite you to submit a revised version of the manuscript that addresses the points raised during the review process.

We look forward to receiving your revised manuscript.

Kind regards,

Alan Ruddock

Academic Editor

PLOS ONE

Journal Requirements:

Reviewers' comments:

Reviewer's Responses to Questions

**Comments to the Author**

1. If the authors have adequately addressed your comments raised in a previous round of review and you feel that this manuscript is now acceptable for publication, you may indicate that here to bypass the “Comments to the Author” section, enter your conflict of interest statement in the “Confidential to Editor” section, and submit your "Accept" recommendation.

Reviewer #1: (No Response)

Reviewer #2: All comments have been addressed

2. Is the manuscript technically sound, and do the data support the conclusions?

Reviewer #1: Yes

Reviewer #2: Yes

3. Has the statistical analysis been performed appropriately and rigorously? 

Reviewer #1: Yes

Reviewer #2: Yes

4. Have the authors made all data underlying the findings in their manuscript fully available?

Reviewer #1: Yes

Reviewer #2: Yes

5. Is the manuscript presented in an intelligible fashion and written in standard English?

Reviewer #1: Yes

Reviewer #2: Yes

6. Review Comments to the Author

Reviewer #1: Thank you for your work in revising this manuscript. I am satisfied with the responses to my previous comments. However, since PLOS ONE does not copyedit manuscripts, I gave a bit more attention to grammar and writing style and have made some suggestions in the attached document for you to consider.

Reviewer #2: The revised version of the article adequately addresses all issues that I previously raised. Introduction and Discussion sectior are written more clearly now.

7. PLOS authors have the option to publish the peer review history of their article (what does this mean?). If published, this will include your full peer review and any attached files.

Reviewer #1: No

Reviewer #2: No

---

## [Author Response · Author response to Decision Letter 1]

15 Dec 2021

Response to Reviewers

Dear Mr. Alan Ruddick,

Thank you for the opportunity to submit a revised version of the manuscript titled Progression in training volume and perceived psychological and physiological training distress in Norwegian student athletes to PLOS ONE. We appreciate the time and effort you and the reviewers have dedicated to providing valuable feedback on the submitted manuscript. We have addressed the raised concerns and hope our responses and revisions will meet your expectations. 

Below is a point-by-point response to the academic editor and the reviewers' comments and concerns. All changes in the manuscript are marked up with red colour.

Journal requirements

Response to the academic editor:

We have double-checked the reference list to ensure that it is complete and correct. Some changes were made and are listed below.

1. Reference 5 is exchanged to the following reference:

Soligard T, Schwellnus M, Alonso J-M, Bahr R, Clarsen B, Dijkstra HP, et al. How much is too much?(Part 1) International Olympic Committee consensus statement on load in sport and risk of injury. Br J Sports Med. 2016;50(17):1030-41.

This reference is part 1 of the reference we removed (part 2). 

2. Reference 1 was listed three times in the reference list. We have adjusted the reference, and it is now listed once. 

3. Reference 49 has been removed due to other more relevant references used in the same sentence (Please see line 354).

4. Reference 6 was listed twice in the reference list. We have adjusted the reference, and it is now listed once.

Reviewer #1

Review Comments to the Author

Thank you for your work in revising this manuscript. I am satisfied with the responses to my previous comments. However, since PLOS ONE does not copyedit manuscripts, I gave a bit more attention to grammar and writing style and have made some suggestions in the attached document for you to consider.

Response to the reviewer:

Thank you for taking the time to assess the manuscript. We have considered your comments and hope our revisions will meet your expectations.

Introduction:

1. Para 1 – I recommend opening with a more general statement and simplifying the rest of the paragraph. My suggested revision is below, which you can take or leave as you see fit. I'm not convinced you need the sentence beginning with "Newly published data."

Response to the reviewer:

We thank the reviewer for pointing this out. We have updated the manuscript with the suggested revision. Please see lines 48-60.

2. Paras 2-3 – I suggest a bit of simplification and reorganization to clarify your reasoning. Regarding Line 64, (first sentence below), I think "determining" would be a better fit than "deciding" here, as "deciding" implies a choice between a set number of options. Although PLOS ONE asks for "Vancouver" referencing and does not give much direction in terms of formatting and writing style, I think it's wise to go with the APA standard of restricting the use of "e.g.," to within parentheses and spelling it out ("For example,") when outside of parentheses. If you go with my suggestions below, please double-check all of the citation numbers.

Response to the reviewer:

Thank you for the suggestions. We have revised the paragraphs (lines 61-83) in line with your suggestions. We have double-checked all of the citation numbers.

3. Lines 106, 109, 114 – Replace "where" with "with." 

Response to the reviewer:

We have made the change and replaced "where" with "with."

4. Lines 109, 112 – Replace "have" vs. "having."

Response to the reviewer:

We have made the changes and replaced "have" with "having."

5. Line 114 – Could reword as "with higher weekly training volume being associated with more training distress."

Response to the reviewer:

Line 109 is adjusted to "with higher weekly training volume being associated with more training distress."

Materials and methods:

1. Lines 142-144 – Update formatting (M ± SD)

Response to the reviewer:

We have made the change in lines 137-138. 

2. Lines 178-180 – Suggested revision: "Cutpoints of 5, 10, and 15 hours of training per week were chosen to ensure relatively equal group sizes"

Response to the reviewer:

We have revised the manuscript following your suggestion. Please see lines 178-179.

3. Line 202 – It would be of great help to students and those of us who do primarily qualitative research if you could include a reminder here of how to interpret these effect sizes (i.e., what constitutes small, medium, and large).

Response to the reviewer:

We thank the reviewer for the suggestion. Please see lines 203 to 207 for a description of how the effect sizes were interpreted. 

4. Line 206 – "…of the sample are presented…"

Response to the reviewer:

We have adjusted the manuscript. 

Results:

1. Description of perceived psychological and physiological training distress – Did you include the scale for this instrument somewhere? Is it from 1-5? This would be good to include here.

Response to the reviewer:

We agree with the reviewer and have included information about the instrument's scale. Please see lines 143-147.

2. Line 326 – I think it would be stronger to cut the first part of this sentence and begin with "No significant differences…"

Response to the reviewer:

We agree and have made the change.

3. Line 340 – Remove the comma after "sports"

Response to the reviewer:

We have made the change.

Discussion:

1. Lines 519-527 – This paragraph seems out of place and somewhat detracts from an otherwise strong conclusion. I would recommend incorporating this content into the results section or the first paragraph of the discussion section. For example, “No significant differences in weekly training volume were observed for gender, school program, or school level, contradicting hypotheses 1b and 1c. However, in line with hypothesis 1a, results indicated a significant difference in weekly training volume between sport types, with endurance sports having a higher training volume than more technically demanding sports.”

Response to the reviewer:

Thank you for pointing this out. We have incorporated the content into the first paragraph of the discussion section (please see lines 332-340).

Reviewer #2

Review Comments to the Author 

The revised version of the article adequately addresses all issues that I previously raised. Introduction and Discussion section are written more clearly now.

Response to the reviewer:

Thank you very much.

---

## [Editor Report · Decision Letter 2]

6 Jan 2022

PONE-D-21-13902R2Progression in training volume and perceived psychological and physiological training distress in Norwegian student athletes: a cross-sectional studyPLOS ONE

Dear Dr. Nyhus Hagum,

Thank you for submitting your manuscript to PLOS ONE. After careful consideration, we feel that it has merit but does not fully meet PLOS ONE’s publication criteria as it currently stands. Therefore, we invite you to submit a revised version of the manuscript that addresses the points raised during the review process.

Dear authors,  Thank you for taking the time to address the reviewers comments. There are some minor editorial amendments for your consideration before we can accept the manuscript for publication. Please see these below in additional editor comments. 

We look forward to receiving your revised manuscript.

Kind regards,

Dr Alan Ruddock

Academic Editor

PLOS ONE

Journal Requirements:

Additional Editor Comments:

L59 – replace high-level with high-standard

L77 – remove capacity from performance capacity – unless you are referring to a physiological concept assessed as a capacity.

I believe your hypotheses would be better written as research questions as these are not in the traditional format since you are intending to falsify the null hypothesis using statistical tests.

L128, 149, – remove the ‘level’ and use ‘standard’ throughout.

Can you confirm how you transformed d into r and whether or not you used r squared (note: you state this as person’s r).

L213 –No need to state P < 0.001 as you state the P value in brackets. Please state the actual P value throughout rather than P < 0.001 etc.

L215 – lower volume – please check throughout and change to less volume.

L216 – Please add units to your mean values throughout. Please add the SD for these means.

Table 1 – Please consider the use of decimal age – E.g. total = 17.29 or 17 years and 105 days or 3.5 months. I think with young adults it is important to state age as clearly as possible.

Table 2 – M, SD and 95% CI needs units.

Line 247 – Partial eta squared first introduced. Need to state this assessment in statistical analysis.

L247 – Larger training volume rather than higher.

Table 3 – see comments for table 2 – in the discussion you refer to a qualitative descriptor. Please add this into the table.

L257 – Different is tautological – if there are dimensions they must be different.

L257 to 259 – This is a description of the data in the table and is repetitive. Please remove.

L263 – Results from multivariate analysis of variance – please specific what this relates to.

L265 and within this section – inconsistent use of significant figures please 2 decimal places

L274 – please remove as hypothesised and present the results only.

L281 – inconsistent use of parentheses to illustrate variables in MANOVA, please remove.

Table 4 would be considered raw data – please consider whether the presentation of this data is essential to the manuscript – there is no explicit reference to table 4 in the discussion.

L299 – units required for ‘group centroids’ (please also see comment on figure 2).

Table 5 – Please consider whether or not the information contained within the table is essential to the manuscript.

Results – please pay attention to the use of the terms highest and lowest throughout

L321 – space required between equal to and greater than sign and 15

L330 – start discussion with the aims of the research, follow with significant findings of this investigation. The use of hypotheses at this point is confusing, please see earlier comment regarding hypotheses.

L338 – this is a conclusion. Please save this for the end of the manuscript after you have created a narrative for your findings.

L351 – replace hyphen with ‘to’ – as in 15 to 18.

Figure 1 – please rename y-axis label as Weekly Training Volume (hours), use integers of 1 hour. School level should be replaced with school year and axis labels as ‘first’, ‘second’ and ‘third’. You also need to use lines that differentiate between the sports – they are all quite similar here. Consider the use of different markers. Your means also need to be accompanied by either SD’s or CI’s.

L369 – please explore the aspect of time of data collection in relation to your data more. Might your data be compromised by this?

L379 – the average could be the mean, mode or median please qualify

L382 – please include additional evidence in your assertion that athletes might be overreaching – distress alone is not sufficient evidence for this

L386 – Here there is little specific relationship to the findings within your study. Please provide more context relating to your study. L393 appears to relate more to speculative practical recommendations rather than a discussion of your findings.

L401 to 420 – this part of the discussion is too statistically orientated, please simply and keep to the main findings of gender differences.

L416 – replace ‘great’ with ‘strong’

L425 – replace higher fatigue with ‘greater’ fatigue

L433 – please be more specific about ‘changes in shape’

L436 – increased muscle mass, larger and longer bones lead to improvements in strength. Please state the subsequent improvements in aerobic capability due an increase in Hb and link central cardiovascular adaptations due to growth.

L470 – as previous section the first paragraph is too statistically dominant please simplify to improve readability and interpretation.

Figure 2 – the use of centroid’s or at least the term is uncommon in sport and exercise science. In your methods section or in your caption to figure 2 please provide an explanation of a centroid in lay terms and the rationale for the use of this term. I think you need to explain the practical importance of this figure in more detail to help the reader understand the message you are trying to convey. In the discussion you make reference to training distress but this is not clear from your figure.
---

## [Author Response · Author response to Decision Letter 2]

21 Jan 2022

Response to Reviewers

Journal requirements

Response to the academic editor:

We have double-checked the reference list to ensure that it is complete and correct.

Additional Editor Comments:

 L59 – replace high-level with high-standard

Response: replaced.

 L77 – remove capacity from performance capacity – unless you are referring to a physiological concept assessed as a capacity.

Response: correct, we are referring to the physiological performance capacity. Adjusted to: "…the necessary adjustments to optimize the physiological performance capacity…

 I believe your hypotheses would be better written as research questions as these are not in the traditional format since you are intending to falsify the null hypothesis using statistical tests.

Response: we agree with the feedback considering that this study is primarily exploratory. We have changed all hypotheses to research questions. Thank you.

 L128, 149, – remove the 'level' and use 'standard' throughout.

Response: Top-level has been changed to top-standard throughout the manuscript.

 Can you confirm how you transformed d into r and whether or not you used r squared (note: you state this as person's r).

Response: Cohen's d (standardized mean difference) and r2 (variance-accounted-for) are related and can be converted into one another. In this study the Cohen's approximate conversion formula was used, then the r multiplied to the power of 2:

r=d/√(d^2+4)

Furthermore, this conversion was conducted to measure the relationship between the variables and the "variance-accounted-for" between variables. The paragraph was rewritten as follow:

"Cohen's d effect sizes were converted to Person's r using Cohen's approximate conversion formula to measure the relationship between variables, and r were then multiplied to the power of 2 (i.e., r2) to be able to estimate the "variance-accounted-for" between variable (Henson, 2006). The relationships between the variables were interpreted based on the guidelines proposed by Funder and Ozer (2019), where an r of 0.05 indicated a very small relationship; an r of 0.10 indicated a small relationship; an r of 0.20 indicated a medium relationship; an r of 0.30 indicated a large relationship; and an r of ≥ 0.40 indicated a very large relationship."

 L213 –No need to state P < 0.001 as you state the P value in brackets. Please state the actual P value throughout rather than P < 0.001 etc.

Response: The P-value in L213 was removed since the P-value was stated in brackets. However, according to the PLOSONE statistical guidelines which state: "P-values. Report exact p-values for all values greater than or equal to 0.001. P-values less than 0.001 may be expressed as p < 0.001, or as exponentials in studies of genetic associations." Hence, we reported the P-values accordingly. All changes are marked with track change. 

 L215 – lower volume – please check throughout and change to less volume.

Response: corrected throughout the manuscript.

 L216 – Please add units to your mean values throughout. Please add the SD for these means.

Response: We have included units, mean values and SD. Please see lines 216-227.

 Table 1 – Please consider the use of decimal age – E.g. total = 17.29 or 17 years and 105 days or 3.5 months. I think with young adults it is important to state age as clearly as possible.

Response: We agree and have adjusted to years ± months. 

 Table 2 – M, SD and 95% CI needs units.

Response: hours (h) are included in Table 2.

 Line 247 – Partial eta squared first introduced. Need to state this assessment in statistical analysis.

Response: The assessment is stated in statistical analysis: Partial eta squared (ηp2) was used to determine the effect size and were interpreted as 0.01 = small, 0.06 = medium, or 0.14 = large (Cohen, 1988, p. 368). Please see lines 166-167. 

 L247 – Larger training volume rather than higher.

Response: corrected throughout the manuscript.

 Table 3 – see comments for table 2 – in the discussion you refer to a qualitative descriptor. Please add this into the table.

Response: MTDS-N score is included in Table 3. A qualitative descriptor was also added below Table 3: a The mean score of the MTDS-N, ranging between 1–5, where 1 = never/ not at all, 2 = almost never/ a little, 3 = sometimes/ moderately, 4 = fairly often/ quite a bit, and 5 = very often/extremely.

 L257 – Different is tautological – if there are dimensions they must be different.

Response: the sentence was removed.

 L257 to 259 – This is a description of the data in the table and is repetitive. Please remove.

Response: the sentence was removed.

 L263 – Results from multivariate analysis of variance – please specific what this relates to.

Response: the subtitle is changed to "The effect of training volume, gender, school level, sport types, and school program on the combined characteristics of training distress".

 L265 and within this section – inconsistent use of significant figures please 2 decimal places

Response: corrected.

 L274 – please remove as hypothesised and present the results only.

Response: corrected.

 L281 – inconsistent use of parentheses to illustrate variables in MANOVA, please remove.

Response: corrected.

 Table 4 would be considered raw data – please consider whether the presentation of this data is essential to the manuscript – there is no explicit reference to table 4 in the discussion.

Response: Thank you for this observation. We have now referred to Table 4 in the discussion (L417 and L491). The DDA and the results in Table 5 are based on the results shown in Table 4. Hence, we think it is essential to the manuscript. 

 L299 – units required for 'group centroids' (please also see comment on figure 2).

Response: the manuscript is adjusted with the following text: “Female student athletes reported higher composite variable means (i.e., training distress) (0.33 ± 1.05; CI = 0.21, 0.45) than males (-0.32 ± 0.95; CI = -0.42, -0.21)”. 

 Table 5 – Please consider whether or not the information contained within the table is essential to the manuscript.

Response: we believe the information is essential to the manuscript. We refer to the table in the discussion (L421, L433, L492). The table gives a good overview of the contribution of each outcome variable.

 Results – please pay attention to the use of the terms highest and lowest throughout

Response: we have read the manuscript several times and believe that the wordings were carefully checked.

 L321 – space required between equal to and greater than sign and 15

Response: corrected.

 L330 – start discussion with the aims of the research, follow with significant findings of this investigation. The use of hypotheses at this point is confusing, please see earlier comment regarding hypotheses.

Response: in line with former comments regarding hypotheses, the manuscript was adjusted accordingly in several sections of the manuscript. All changes are marked with track change.

 L338 – this is a conclusion. Please save this for the end of the manuscript after you have created a narrative for your findings.

Response: agreed and removed from this paragraph and moved to the conclusion.

 L351 – replace hyphen with 'to' – as in 15 to 18.

Response: corrected.

 Figure 1 – please rename y-axis label as Weekly Training Volume (hours), use integers of 1 hour. School level should be replaced with school year and axis labels as 'first', 'second' and 'third'. You also need to use lines that differentiate between the sports – they are all quite similar here. Consider the use of different markers. Your means also need to be accompanied by either SD's or CI's.

Response: the figure is adjusted. Further, we have included markers to better differentiate between sports. The means are accompanied by SD’s.

 L369 – please explore the aspect of time of data collection in relation to your data more. Might your data be compromised by this?

Response: we have reformulated the paragraph to: "The periods within a training macrocycle could potentially contribute to explain this finding. It is well known that different sports have different competition periods within a training macrocycle, which might have influenced the reported training volume. Hence, athletes who are in the competition season likely have less volumes with higher intensities. In comparison, athletes in the preparatory phase may have larger volumes with lower intensities where the focus is more on technical skills and the development of the general physical base".

 L379 – the average could be the mean, mode or median please qualify

Response: corrected. Mean ± SD.

 L382 – please include additional evidence in your assertion that athletes might be overreaching – distress alone is not sufficient evidence for this

Response: additional evidence was added to the text: "Conversely, a study of 17 elite Australian rowers demonstrated a decline in performance in 5 km rowing combined by altered pacing strategy, suggesting an increase in fatigue while simultaneously the total training distress scores increased significantly following four weeks of intensified training, suggesting that the athletes may have reached a short-term performance decrements accompanied by psychological and physiological symptoms including mood disturbance (Woods et al., 2017). Similar results have been found in fourteen male cyclists during a six-week training program, where increased training distress was significantly associated with increased training load (~150% of regular training load) (Woods et al., 2018)."

 L386 – Here there is little specific relationship to the findings within your study. Please provide more context relating to your study. L393 appears to relate more to speculative practical recommendations rather than a discussion of your findings.

Response: this section was improved by relating the results from these studies to the results of this study. E.g., "Comparing the findings from these studies (54, 55) to the findings from this study suggest that the training load was not sufficient enough for the student athletes to reach high training distress indicated by the observed "little" to "moderate" amount of training distress scores. Furthermore, it has been demonstrated that those experiencing positive training adaptations are more likely to score highly on negative dimensions included in MTDS (56), which was not observed in this study. Such results suggest that the training load must be high enough to cause stress in order to induce the desired training adaptation."

 L401 to 420 – this part of the discussion is too statistically orientated, please simply and keep to the main findings of gender differences.

Response: this part was rewritten, maintaining the findings of this study and reducing the statistical terms used.

 L416 – replace 'great' with 'strong'

Response: corrected.

 L425 – replace higher fatigue with 'greater' fatigue

Response: corrected.

 L433 – please be more specific about 'changes in shape'

Response: corrected to "human body shape".

 L436 – increased muscle mass, larger and longer bones lead to improvements in strength. Please state the subsequent improvements in aerobic capability due an increase in Hb and link central cardiovascular adaptations due to growth.

Response: we have adjusted the manuscript with the following text: Conversely, males typically experience physical performance improvements during adolescence. Larger and stronger bones and increased muscle mass lead to improvements in strength (Handelsman, 2017). In addition, increased circulation hemoglobin and greater muscle mass make a significant contribution to increases in peak oxygen uptake (Armstrong & Welsman, 2019).

 L470 – as previous section the first paragraph is too statistically dominant please simplify to improve readability and interpretation.

Response: this part was rewritten, maintaining the findings of this study and reducing the statistical terms used.

 Figure 2 – the use of centroid's or at least the term is uncommon in sport and exercise science. In your methods section or in your caption to figure 2 please provide an explanation of a centroid in lay terms and the rationale for the use of this term. I think you need to explain the practical importance of this figure in more detail to help the reader understand the message you are trying to convey. In the discussion you make reference to training distress but this is not clear from your figure.

Response: the word centroid has been replaced with "composite variable means" throughout the manuscript. We have explained that training distress is the composite variable. We have changed the caption in Fig 2 to: "Linear discriminant function plot showing the interaction of weekly training volume by school year and how the training volume groups separate.” In addition, we have included the following fig legend: “The figure shows the means of each training volume group on the composite outcome variable that was created from the observed variables (i.e., training distress). To facilitate the interpretation of the figure, both the rs and the standardized coefficients from Table 5 could be examined.”

References

Armstrong, N., & Welsman, J. (2019). Development of peak oxygen uptake from 11–16 years determined using both treadmill and cycle ergometry. European journal of applied physiology, 119(3), 801-812. 

Cohen, J. (1988). Statistical power analysis for the behavioral sciences (2nd ed.). Laurence Erlbaum. 

Funder, D. C., & Ozer, D. J. (2019). Evaluating effect size in psychological research: Sense and nonsense. Advances in Methods Practices in Psychological Science, 2(2), 156-168. 

Handelsman, D. J. (2017). Sex differences in athletic performance emerge coinciding with the onset of male puberty. Clinical endocrinology, 87(1), 68-72. 

Henson, R. K. (2006). Effect-size measures and meta-analytic thinking in counseling psychology research. The Counseling Psychologist, 34(5), 601-629. 

Woods, A. L., Garvican-Lewis, L. A., Lundy, B., Rice, A. J., & Thompson, K. G. (2017). New approaches to determine fatigue in elite athletes during intensified training: Resting metabolic rate and pacing profile. PLoS One, 12(3), e0173807. 

Woods, A. L., Rice, A. J., Garvican-Lewis, L. A., Wallett, A. M., Lundy, B., Rogers, M. A., Welvaert, M., Halson, S., McKune, A., & Thompson, K. G. J. P. o. (2018). The effects of intensified training on resting metabolic rate (RMR), body composition and performance in trained cyclists. 13(2).

---

## [Editor Report · Decision Letter 3]

24 Jan 2022

Progression in training volume and perceived psychological and physiological training distress in Norwegian student athletes: a cross-sectional study

PONE-D-21-13902R3

Dear Dr. Nyhus Hagum,

We’re pleased to inform you that your manuscript has been judged scientifically suitable for publication and will be formally accepted for publication once it meets all outstanding technical requirements.

Kind regards,

Alan Ruddock

Academic Editor

PLOS ONE
---

## [Editor Report · Acceptance letter]

27 Jan 2022

PONE-D-21-13902R3 

Progression in training volume and perceived psychological and physiological training distress in Norwegian student athletes: a cross-sectional study 

Dear Dr. Nyhus Hagum:

I'm pleased to inform you that your manuscript has been deemed suitable for publication in PLOS ONE. Congratulations! Your manuscript is now with our production department. 

Kind regards, 

on behalf of

Dr. Alan Ruddock 

Academic Editor

PLOS ONE